# Connectivity of Fennoscandian Shield terrestrial deep biosphere microbiomes with surface communities

George Westmeijer [1✉], Maliheh Mehrshad[2], Stephanie Turner[1], Linda Alakangas[3], Varvara Sachpazidou[4], Carina Bunse[1,5], Jarone Pinhassi [1], Marcelo Ketzer [4], Mats Åström[4], Stefan Bertilsson [2] & Mark Dopson [1]

The deep biosphere is an energy constrained ecosystem yet fosters diverse microbial communities that are key in biogeochemical cycling. Whether microbial communities in deep biosphere groundwaters are shaped by infiltration of allochthonous surface microorganisms or the evolution of autochthonous species remains unresolved. In this study, 16S rRNA gene amplicon analyses showed that few groups of surface microbes infiltrated deep biosphere groundwaters at the Äspö Hard Rock Laboratory, Sweden, but that such populations constituted up to 49% of the microbial abundance. The dominant persisting phyla included Patescibacteria, Proteobacteria, and Epsilonbacteraeota. Despite the hydrological connection of the Baltic Sea with the studied groundwaters, infiltrating microbes predominantly originated from deep soil groundwater. Most deep biosphere groundwater populations lacked surface representatives, suggesting that they have evolved from ancient autochthonous populations. We propose that deep biosphere groundwater communities in the Fennoscandian Shield consist of selected infiltrated and indigenous populations adapted to the prevailing conditions.

[1] Centre for Ecology and Evolution in Microbial Model Systems (EEMiS), Linnaeus University, Stuvaregatan 4, 39 231 Kalmar, Sweden. [2] Department of Aquatic Sciences and Assessment, Swedish University of Agricultural Sciences, P.O. Box 7050, 75 007 Uppsala, Sweden. [3] Swedish Nuclear Fuel and Waste Management Co (SKB), 57 229 Oskarshamn, Sweden. [4] Department of Biology and Environmental Sciences, Linnaeus University, 39 231 Kalmar, Sweden. [5] Helmholtz-Institute for Functional Marine Biodiversity at the University of Oldenburg (HIFMB), 26129 Oldenburg, Germany. ✉email: george.westmeijer@lnu.se

The terrestrial deep biosphere is temporally and spatially separated from the Earth's photosynthesis-driven surface, yet accommodates a diverse microbial community with an estimated number of 2 to $6 \times 10^{29}$ cells[1], that is key in driving the Earth's biogeochemical cycles[2–5]. Despite limitations imposed by the predominantly extreme low carbon and energy conditions, subsurface microbial communities are alive[6], active[7–9], and metabolically versatile; exhibiting energy conservation strategies that include nitrate and sulfate reduction, fermentation, methanogenesis, and acetogenesis[10–12]. Life in the terrestrial subsurface can be dictated by infiltration of organic carbon[13], while communities at greater depths are largely separated from surface input and are sustained by alternative sources of energy such as the abiotically produced "geogases" carbon dioxide and hydrogen[13–15] and iron and manganese-rich minerals[16]. However, the limited study sites compared to the vast volume of the deep biosphere biome results in largely undescribed communities including how the microbial assemblages are shaped.

Deep subsurface microbial communities exhibit distinct traits to cope with low energy flux, including small cell size[10], streamlined genomes[17,18], and a high functional interactivity such as syntrophy[3,19,20]. For instance, Patescibacteria are suggested to be ubiquitous in shallow groundwaters due to their ease of mobilization from soils, small genomes, and ability to survive in low carbon and energy conditions[21,22]. In contrast to the paucity of deep terrestrial biosphere studies, the marine sub-seafloor sediment microbial diversity has been shown to decrease with depth and consequently age of the sediment[23–25] in response to gradients in both temperature and the availability of organic carbon[26]. Starnawski et al.[27] also identified a sub-seafloor community derived by selection of a small group of surface sediment taxa with the ability to persist under severe energy limitation for >5000 years. In addition, analysis of marine sulfate reducing microbes has shown that the sediment surface community influences sub-seafloor populations through the process of species sorting whereby the geochemical conditions shape the microbial communities by favoring distinct populations[28]. Additional work on Indian Ocean and up to 1.3 Ma old Bering Sea sediments obtained from depths down to 332 m below the seafloor shows community dependence on the relative abundance of the population in the near-seafloor sediment[29]. Yet, for the terrestrial deep subsurface, the question remains whether surface microbes infiltrate the deep biosphere and possess traits that allow them to survive in this predominantly low energy and nutrient poor ecosystem or if an indigenous deep groundwater community has developed that is independent of the surface world.

Previous studies have explored the microbial diversity of surface waters[30], and benthic waters plus sediments[31–33] in the south-western part of the Baltic Sea region. Microbial communities in the deep biosphere were investigated in fractures intersected by boreholes emanating from the Äspö Hard Rock Laboratory (HRL) that is built partly under the land and partly under the sea. Äspö HRL, operated by the Swedish Nuclear Fuel and Waste Management Company, is a 3.6 km long tunnel extending 460 m below sea level into ~1.8 Ga years old fractured bedrock consisting of Fennoscandian Shield Paleoproterozoic granitoids[34–36] that bear groundwaters with contrasting depths, chemical compositions, and residence times[10,13,37]. Water from the terrestrial landscape and the Baltic Sea are transported deep into the bedrock via vertical to subvertical fractures and thus, provide an ideal situation to investigate how the microbial communities in the deep biosphere groundwaters are influenced by different surface communities[37–39]. The groundwaters are distinguished according to their origin with "meteoric groundwater" derived from the land surface, "modern marine groundwater" from the overlying Baltic Sea, and "old saline

groundwater" thought to be completely separated from surface energy influx[10,13,39–41]. Many microbiological studies have been carried out at Äspö HRL including 'omics' investigations into community structure and dynamics[10,13,37]. These microbial communities survive in the deep biosphere by syntrophy and symbiotic associations that alleviates the 'tragedy of the commons'[42] that is aided by biofilm/aggregate formation[43].

Here we set out to elucidate the potential role of infiltration into deep biosphere groundwaters of surface and near-surface microbes for the structuring of the deep microbiota. 16S rRNA gene amplicon datasets from nearby aquatic, sediment, soil, and terrestrial deep biosphere environments were collected to unravel potential similarities between their microbial communities. We hypothesized that the fixed niches harboring a common core microbial deep biosphere biome[42] drive species sorting of surface microbes, thereby resulting in reduced diversities.

## Results and discussion

**Geochemistry and water flow**. The meteoric and the deeper modern marine and old saline groundwaters have different characteristics including the concentration of dissolved organic carbon (DOC), chloride concentration, and $\delta^{18}O$ values (Fig. 1 plus S1 for additional parameters). The shallowest sampled bedrock fracture carried a groundwater below the surface of the Äspö Island that was classified as containing 80% meteoric water based upon $\delta^{18}O$ values, chloride concentration, low magnesium plus sulfate concentrations, and high DOC content. This groundwater constituted precipitation that originally soaked into the soil groundwater and was transported downward to reach the meteoric groundwater after eight months to a year[39,44]. Chloride concentrations and $\delta^{18}O$ values of the modern marine groundwater ranged from 2.06 to 4.10 g L$^{-1}$ and −7.3 to −9.2‰ relative to Standard Mean Ocean Water, respectively, that were similar to the 3.38 g L$^{-1}$ and −6‰ values, respectively reported in the Baltic Sea by Mathurin et al.[39]. These data supported the modern marine groundwater being infiltrated by Baltic Sea water with a residence time of several years up to 20 years[39]. Additional evidence for the infiltration is that the modern marine groundwater was connected with the overlying Baltic Sea via extensive vertical to subvertical fractures, leading to drawdown of marine water[45]. Finally, old saline groundwater had chloride concentrations in the range of 12.0 to 16.2 g L$^{-1}$ and $\delta^{18}O$ values in the range of −11.3 to −12.4‰. The low $\delta^{18}O$ values in particular revealed a high degree of separation from other groundwater types and hence, residence times that would extend to thousands of years or more[45]. The old saline groundwater likely originated as a marine water that underwent substantial changes in chemistry over geological time scales when in contact with bedrock surfaces and secondary minerals on the fracture walls[38]. However, it cannot be ruled out that marginal mixing between the sampled fractures occurred, such that the old saline groundwater could contain a minor fraction of modern marine groundwater and vice versa. This can be regarded as a minor disturbance that does not substantially affect the overall results and interpretations.

**16S rRNA gene amplicon sequencing**. 16S rRNA gene amplicon sequences (totaling 214 samples) were analyzed from a variety of environments including surface and benthic seawater; upper (depth 0–1 cm) and lower sediments (6 and 20 cm); terrestrial upper (2–3 m) and lower (5 m) soil groundwaters; and meteoric, modern marine, and old saline deep groundwaters residing in bedrock fractures at depths between 70 and 460 m below the surface. The samples from the surface seawater, lower sediment, upper and lower soil groundwaters, and meteoric groundwaters were sequenced for this study while the data from the remaining

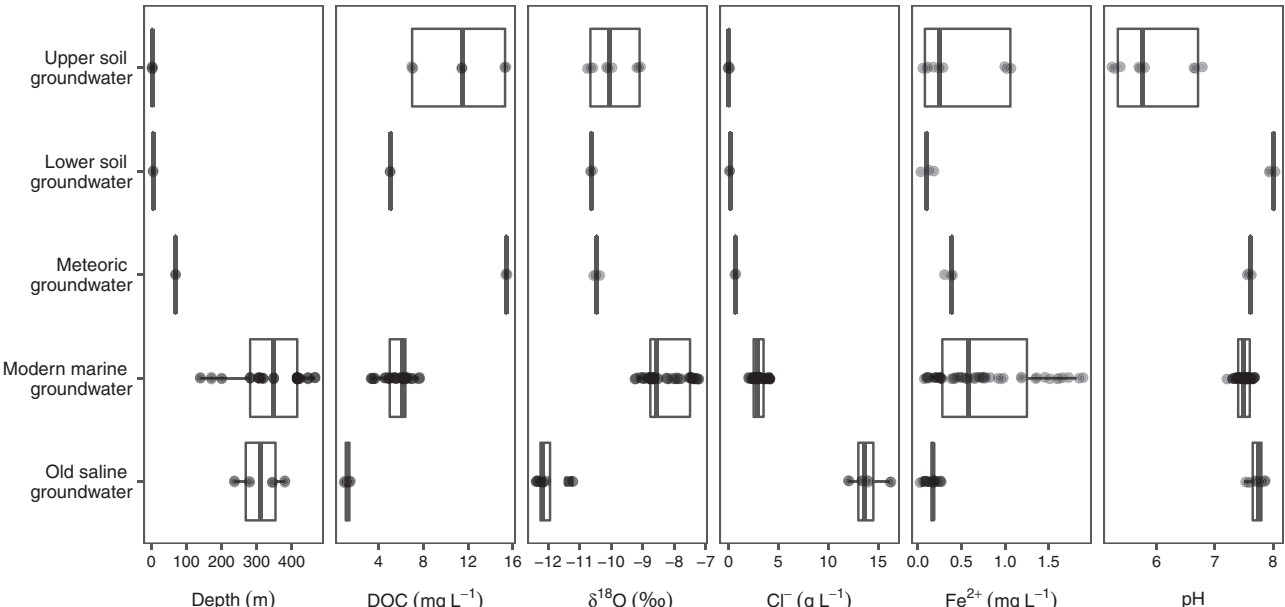

**Fig. 1 Boxplot depicting depth and chemistry of the studied soil plus deep biosphere groundwaters.** DOC; dissolved organic carbon. $\delta^{18}O$ is the $^{18}O/^{16}O$ ratio relative to Standard Mean Ocean Water, expressed in parts per thousand (‰). The boxes are formed by the first and third quartiles while the whiskers extend to 1.5 times the inter-quartile range. Additional chemical parameters including strontium and manganese concentrations are shown in Fig. S1.

environments have been previously published. An overview figure of the various sampling sites is provided in Fig. 2, with sequencing data repository references in Table S1, and sampling details in Table S2. In total 18 million sequencing reads were included in this study, encompassing 48.7 thousand amplicon sequence variants (ASVs). On average, each sample contained $83 \pm 72$ thousand sequencing reads (mean ± standard deviation). The upper sediment samples had the lowest sequencing depth with $34 \pm 19$ thousand reads on average per sample ($n = 19$) while the old saline groundwater samples had $108 \pm 72$ thousand reads ($n = 15$; details provided in Table S3). The rarefaction curves depicting the relationship between sequencing depth and ASV count are asymptotic for nearly all samples, indicating a sufficient sequencing effort for microbial diversity estimates (Fig. S2).

**Microbial community description.** On phylum level, the Baltic surface seawater community had a similar composition to the Baltic benthic seawater community. Likewise, the composition of sediment communities (upper and lower) resembled each other along with the soil versus deep biosphere groundwaters communities (Fig. 2). The Actinobacteria and Cyanobacteria were the most abundant phyla in the Baltic surface seawater community with an abundance of 30 and 23%, respectively. In contrast, both upper and lower sediment communities were characterized by a high abundance of Proteobacteria (47 and 37%, respectively), Bacteroidetes (15 and 8%), and Chloroflexi (5 and 16%). Finally, both soil and deep biosphere groundwater communities were characterized by a high abundance of Proteobacteria and Patescibacteria with a combined abundance above 50% that increased up to 74% in the upper soil groundwater. The high abundance of the Patescibacteria clearly distinguished the soil plus deep biosphere groundwaters from the sediment plus seawater communities (Fig. 2) and is consistent with the Patescibacteria being abundant in shallow aquifers[21,46]. The Patescibacteria dominated the upper soil and meteoric groundwaters (49 and 43%) after which the abundance of this clade decreased with depth in the modern marine (28%) and old saline (21%) groundwaters. Abundant representatives from the Patescibacteria included

"*Candidatus* Falkowbacteria" and "*Ca.* Kaiserbacteria" (Fig. S3) that both constituted more than 10% of the shallow soil and modern marine groundwater communities. The persistence of this clade in subsurface groundwaters was likely due to its capacity to maintain growth under low energy conditions[21,46]. Within the Proteobacteria, the Betaproteobacteriales and the Campylobacterales orders were mainly responsible for the abundance of this phylum in the modern marine and old saline groundwaters (Fig. S3). On genus level, *Sulfurimonas* and *Thiobacillus* were abundant representatives of the Proteobacteria, together comprising 27 and 17% of the microbial abundance in the modern marine and old saline groundwaters, respectively. Although Cyanobacteria are reported to be present in the deep biosphere[47,48], this phylum was only scarcely present in the meteoric (0.4%) and modern marine (0.02%) groundwaters and was not detected in the old saline groundwater.

Alpha diversity, according to the Shannon index, peaked in the lower sediment ($6.7 \pm 0.60$; Fig. 3) and upper soil groundwater ($6.6 \pm 1.1$) before decreasing with depth to $3.9 \pm 0.64$ in the old saline groundwater community. For example, the alpha diversity in the lower sediment community was statistically higher than in the modern marine groundwater community (Tukey HSD's $p$ value $= 6.3e^{-3}$, all pairwise tests in Table S4). However, the lower sediment community alpha diversity was higher compared to the upper sediment (albeit insignificant, $p = 0.22$) that contrasts with the general notion that diversity decreases with sediment depth[27,49]. This incongruence was potentially caused by sampling a larger part of the deeper sediment column (i.e. top 1 cm for upper sediment compared to 6 plus 20 cm depth for lower sediment), thereby capturing more fine-scale variation e.g., local diversity hotspots in redox transition zones[50], that would positively affect the overall diversity. Within the groundwaters, a negative correlation between alpha diversity and depth was observed (Pearson's $r = -0.73$, $p = 2.7e^{-13}$; Fig. 3) and also between diversity and chloride content ($r = -0.33$, $p = 0.005$) while the correlation of alpha diversity with DOC was positive ($r = 0.60$, $p = 1.1e^{-7}$). In addition, depth correlated with DOC ($r = -0.63$, $p = 1.1e^{-3}$) that in general showed the deeper

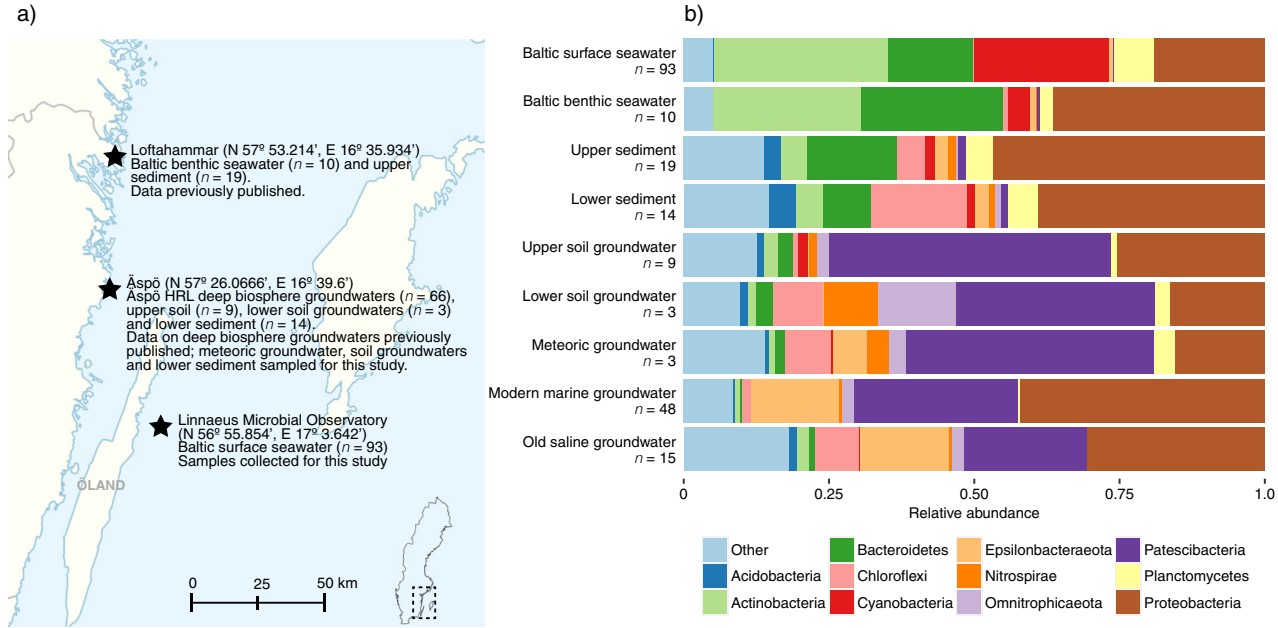

**Fig. 2 Overview of sampling sites and their microbial communities. a** Map of south-east Sweden with the sampling sites marked as a star. **b** Bar plot depicting the 11 most abundant phyla over all the environments with the remaining phyla grouped as "Other". The number of replicates within each environment is displayed underneath each bar label. The environments are ordered according to increasing depth although they do not represent a physical continuum due to the multiple sampling sites as shown in (**a**).

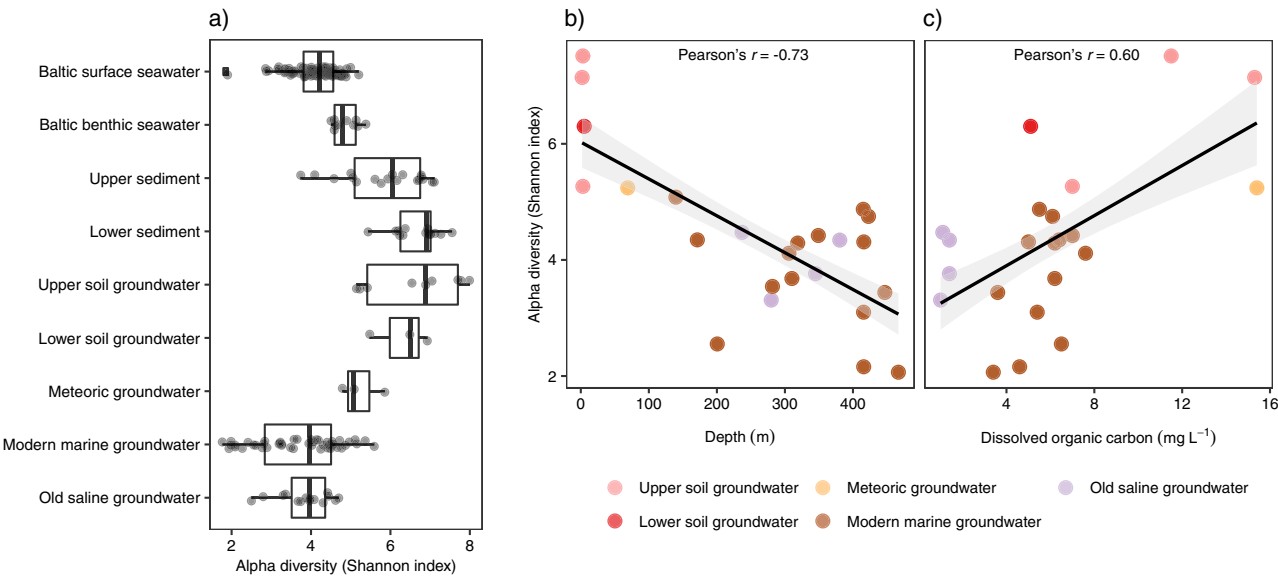

**Fig. 3 Alpha diversity microbial communities. a** Boxplot combined with a dot plot. The box is formed by the first and third quartiles while the whiskers extend to 1.5 times the inter-quartile range. The data points in the Baltic surface seawater left of the whisker indicates outliers. The environments are ordered according to increasing depth although they do not represent a physical continuum due to the multiple sampling sites as shown in Fig. 2a. Scatterplots depicting the relation of alpha diversity with depth (**b**) and with dissolved organic carbon (**c**). Shading in **b**, **c** represents a 95% confidence interval.

groundwaters contained less DOC and have reduced diversities. These correlations help explain why the old saline groundwater had a reduced diversity compared to e.g., the meteoric groundwater as the former is at greater depth and has a very low organic carbon content (Fig. 1). The ordination plot (Fig. 4) revealed a high dissimilarity between deep biosphere groundwater communities and Baltic surface and benthic, upper and lower soil, plus upper and lower sediment microbial communities, which was confirmed by statistical testing (Table S5). The modern marine groundwater clusters with the old saline groundwater (Fig. S4)

while in contrast, the meteoric groundwater sat alone between the other deep biosphere groundwaters and the lower soil groundwater.

The results showed that the deep biosphere groundwaters had a lower alpha diversity than soil groundwaters and sediments. This diversity decreased with depth, retention time from a few years in meteoric to thousands of years in old saline groundwater, and dissolved organic carbon content (Fig. 3). Despite infiltration of the Baltic Sea, the microbial community in the modern marine groundwater did not resemble the community in the Baltic

surface seawater, as illustrated by the composition on phylum level (Fig. 2) and beta diversity analysis (Fig. 4).

**Connectivity of surface microbes with the deep biosphere.** All deep biosphere groundwaters shared at least 8% of their ASVs with the surface and near-surface environments (Fig. 5), supporting the concept that "*Everything is everywhere, but, the environment selects*"[51]. For example, of the 8567 ASVs in the modern marine groundwater, 748 ASVs had a (near) surface representative and these taxa accounted for 49% of the abundance in the modern marine groundwater community. 572

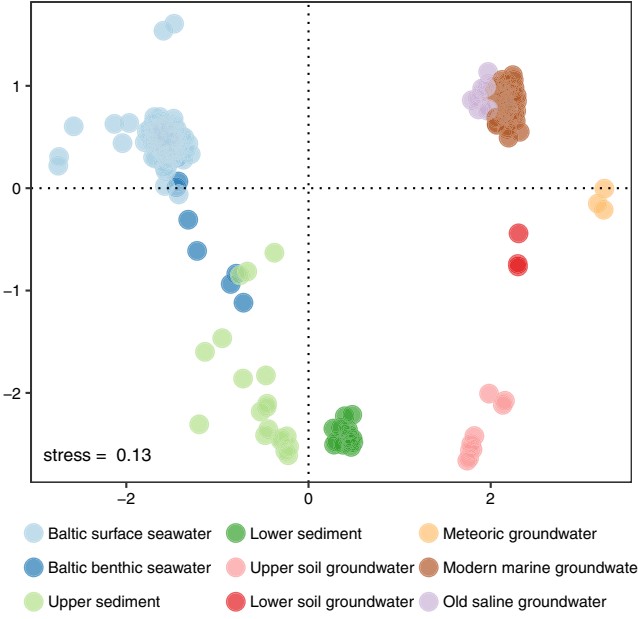

**Fig. 4 NMDS of microbial communities from surface, near-surface, and deep biosphere environments.** The beta diversity was estimated according to the Bray-Curtis dissimilarity index. The clustering was verified by multivariate statistical testing (Table S4).

out of 2836 ASVs in the old saline groundwater had a (near) surface representative that accounted for 36% of this groundwater community (Fig. 5). Despite seawater infiltration of bedrock fractures[37], the number and the abundance of persisting ASVs between the Baltic surface seawater and modern marine groundwater was low (Table S6). Pairwise comparison of the various communities revealed that the lower soil groundwater was the most important source of surface microbes to the deep biosphere groundwaters (Table S6). For example, 227 out of a total of 1660 meteoric ASVs persisted between lower soil and meteoric groundwaters that constituted 22% of the abundance in the meteoric groundwater community. This overlap was dominated by the Patescibacteria, demonstrating that the capacity of ultra-small cells to infiltrate shallow groundwaters from soils[21] also extends to the deep terrestrial biosphere[22]. Potentially, the dominance of this group among the persisting ASVs is facilitated by their episymbiosis on bacterial or archaeal hosts and this lifestyle may also be an adaptation to low-energy environments such as the deep biosphere[52]. In addition to the Patescibacteria, the Proteobacteria and Epsilonbacteraeota phyla were also abundant in the overlapping community (Fig. 5) and interestingly, the Epsilonbacteraeota phylum was only represented by the genera *Sulfurimonas* and *Sulfuricurvum*. The most abundant genera affiliated with the Proteobacteria were *Syntrophus* and *Hydrogenophaga*. Most Proteobacteria and Epsilonbacteraeota likely survived due to the prevalence and importance of sulfate reduction in oligotrophic deep biosphere groundwaters[53,54].

The modern marine groundwater shared 2186 out of its 9315 ASVs with the old saline groundwater and these ASVs comprised 92 and 98% of the abundance in these groundwaters, respectively. This high degree of overlap between both communities is also depicted in Fig. 4 wherein the samples of both environments form a cluster. That the old saline groundwater community was predominantly a subset of the modern marine groundwater community suggested that its diversity was constrained by its geochemistry such as a very low organic carbon content (1–1.4 mg L$^{-1}$), long retention time (up to thousands of years), and very high chloride content (12–16 g L$^{-1}$). In contrast, the modern marine groundwater community shared relatively few

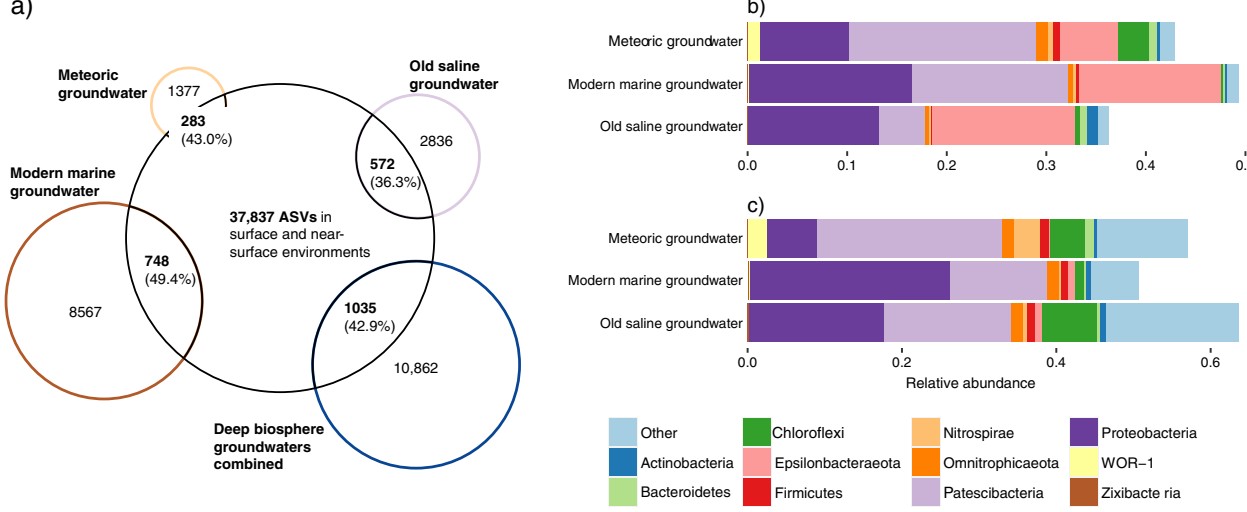

**Fig. 5 Shared ASVs between deep biosphere groundwaters and surface plus near-surface environments. a** Venn diagram depicting the number of shared ASVs between each of the groundwaters and the surface plus near-surface communities with their relative abundance in parentheses. Surface and near-surface is a reference to the seawater, sediment, and soil groundwater samples. **b** The taxonomy of the shared ASVs sorted at the taxonomic level of phylum. **c** The taxonomy of the ASVs that lack a surface or near-surface representative, sorted at the taxonomic level of phylum. For each groundwater, the relative abundance of **b**, **c** sums up to 1. The overlap among the deep biosphere groundwater communities can be found in Table S5.

taxa with e.g., the Baltic benthic seawater and upper sediment communities (Table S6a) but these shared ASVs comprised 34 and 33% of this groundwater's microbial abundance, respectively (Table S6b). Likewise, the old saline groundwater community shared few taxa with e.g., the upper sediment community but these shared ASVs comprise 23% of this groundwater's microbial abundance. Starnawski et al.[27] found a similar pattern with 79 OTUs persisting throughout the marine sediment column that made up to as much as 40–50% of the microbial community in deeper sediment layers while these OTUs represented less than 10% of the total diversity. Although a significant proportion of the diversity in the deep biosphere groundwaters originated from surface infiltration, it is not entirely a subset. This is illustrated in Fig. 5 where the majority of deep biosphere ASVs lack a (near) surface representative and suggests that part of the diversity in the deep biosphere likely constituted a long-term community separated from the surface for many thousands of years. However, it should be noted that these missing surface representatives might not have been captured in the (near) surface samples due to fine-grained spatial heterogeneity in e.g., the soil groundwaters. That only a small number of ASVs from the Baltic surface seawater persisted in deep biosphere groundwaters, despite infiltration of seawater[37], shows that surface microbes unable to survive in the challenging low energy conditions were outcompeted and the cellular components rapidly consumed during nutrient recycling[6].

**Core and accessory deep biosphere taxa.** The deep biosphere groundwaters shared 154 out of a total of 9025 ASVs that were present in all three groundwater types. These ASVs, referred to as the core community, constituted 24, 24, and 14% of the microbial community in the meteoric, modern marine, and old saline groundwaters, respectively of which 52 were affiliated with the Patescibacteria (Fig. S5). Abundant taxa within the Patescibacteria included "*Ca*. Falkowbacteria" in the modern marine groundwater (12%) along with "*Ca*. Levybacteria" (5.7%) and "*Ca*. Kaiserbacteria" (2.1%) in the meteoric groundwater. Nine of these core ASVs were affiliated with the genus *Syntrophus* (class Deltaproteobacteria) of which representatives have been identified in groundwater from a 1.34 km deep former mine[55]. In addition to the core community, the meteoric, modern marine, and old saline groundwaters harbored an accessory community of 1324 (67% of the community), 4895 (6.0%), and 978 (2.1%) unique ASVs, respectively that were not retrieved in any other environment. 29% of the ASVs unique to the meteoric groundwater were affiliated with the Patescibacteria (Fig. S5). The contrast between the three groundwater types, especially between meteoric groundwater and modern marine plus old saline groundwater, suggested a role of the groundwater's differing depth, retention time, and chemistry that favors different microbial populations. Finally, this analysis of deep biosphere groundwaters confirmed the previously reported presence of a core microbiome[42].

**Conclusions.** The results reveal connectivity of surface and near-surface environments with the deep biosphere at the Äspö HRL, allowing infiltration of microbes. Consequently, the deep biosphere is potentially prone to alterations in surface biome microbial communities, such as climate change. Although the number of ASVs persisting among various environments was relatively low, these taxa comprise up to 49% of the deep biosphere microbial community. Despite hydrological connectivity of the Baltic Sea with the majority of the studied deep biosphere groundwaters, the most important source of microbes from the (near) surface was the lower soil groundwater. This supported the concept of species sorting whereby in this case, microbial populations migrate, but the restrictive geochemical conditions in the deep biosphere selects. The

abundant persistent ASVs aligned with the phyla Patescibacteria, Proteobacteria, and Epsilonbacteraeota. Finally, the absence of a surface representative for the majority of abundant deep biosphere taxa suggested that a large portion of the deep biosphere microbiome were from resident autochthonous populations that live isolated from the photosynthetic driven surface.

## Methods

**Published datasets used in this study.** A total of 29 previously published 16S rRNA gene amplicon (V3–V4 region) samples, each sampled in replicates were included in this study (Table S1). Baltic benthic seawater and upper sediment samples were collected using a gravity corer, sampling the top 1 cm layer of sediment and the overlying benthic water[31–33]. Modern marine and old saline groundwaters from the Äspö HRL were collected under in situ pressure using a 0.1 μm filter connected to the respective boreholes as previously described[13]. The microbial communities in Baltic benthic seawater plus upper sediment[31–33] and modern marine plus old saline groundwaters[13] have been previously described.

**Sampling sites and cell capture.** In addition to the published amplicon data, a further 122 samples were collected from various environments (overview provided in Fig. 1 and sampling details in Table S2). Baltic surface seawater samples ($n = 93$) were captured as part of a time-series between October 2011 and December 2013 at the Linnaeus Microbial Observatory (LMO), located 11 km off the northeast coast of Öland using a Ruttner sampler at a depth of 2 m below the water surface[56]. The lower sediment ($n = 14$) was sampled in June 2018 with a gravity corer in a bay near Äspö HRL (Borholmsfjärden) according to Broman et al.[33] except that the sediment core was cut at 6 and 20 cm below the sediment-water interface. To account for spatial heterogeneity, seven sites within Borholmsfjärden were sampled at two depths, yielding 14 samples. With salinity as the major driver for microbial community composition in the Baltic Sea[57], salinity concentrations were measured for the surface seawater samples in LMO (range 6.6–7.6%), the benthic water in Loftahammar (6.5%)[32], and the water overlying the sediment in Borholmsfjärden (range 6.1–6.5%). A one-way ANOVA showed no significant differences in salinity regimes among the three environments ($p = 0.15$). Furthermore, soil groundwaters were sampled in October 2019 using soil tubes available on Äspö Island designated as upper (2–3 m below the surface; three tubes; SSM42, SSM215, & SSM268, $n = 9$) and lower (5 m depth; one tube; SSM22, $n = 3$) soil groundwater. Finally, meteoric groundwater (borehole KR0015B at 70 m depth, $n = 3$) was sampled in October 2019. Both soil and meteoric groundwaters were sampled in triplicates as described for the modern marine and old saline groundwaters in Lopez-Fernandez et al.[13]. Briefly, to avoid contamination from stagnant water in the sampling connections and boreholes, three section volumes were flushed and discarded before connecting a high-pressure filter holder (Millipore) containing a sterile 0.1 μm pore size polyvinylidene fluoride Durapore Membrane filter and the water allowed to pass through the filter. The filters were aseptically collected in cryotubes and immediately frozen in liquid nitrogen for transport to the laboratory where the samples were kept at −80 °C.

The chemistry data was part of SKB's extensive geochemical monitoring program that is publicly available as the Sicada database. For the modern marine and old saline groundwaters, this data has been published in Lopez-Fernandez, et al.[13]. For the meteoric and soil groundwaters, chemistry data was retrieved for this study with the sampling dates (November 2019) as close as possible to the microbiology sampling effort during October 2019.

**DNA extraction, PCR amplification, and sequencing.** DNA was extracted using a phenol-chloroform based method[58] for the Baltic surface seawater samples; the Qiagen DNeasy PowerSoil Kit for the upper and lower sediment samples; and the Qiagen DNeasy PowerWater Kit for both soil and meteoric groundwater samples. Regarding the soil and meteoric groundwater samples, the manufacturer's protocol was followed apart from re-suspending the extracted DNA in 50 μL instead of 100 μL elution buffer. 16S rRNA gene fragment sequences (V3–V4 region) were PCR amplified using the primer set 341F and 805R[57] according to the protocol described in Hugerth et al.[59]. DNA concentration was analyzed using a Qubit 2.0 Fluorometer (Life Technologies) and amplicon specificity by gel electrophoresis. Sequencing was carried out at the Science for Life Laboratory, Sweden on the Illumina MiSeq platform, producing $2 \times 300$ bp paired-end reads[13].

**Bioinformatic analyses.** Raw sequencing reads were processed using the Ampliseq pipeline (v1.2.0)[60] that relied on FastQC (v0.11.8), Cutadapt (v2.8)[61], MultiQC (v1.9)[62], QIIME2 (v2019.10.0)[63], DADA2 (v1.14.1)[64], and the SILVA reference database (v138.1)[65]. Samples with less than 1000 reads were excluded from downstream analysis, thereby removing three from the 96 samples from the Baltic surface seawater environment. To test for DNA extraction kit contaminants that can be an issue in low biomass environments, four negative control DNA extracts were sequenced and processed using identical parameters. One sample yielded a total of 86 ASVs that were removed from the dataset prior to downstream analysis.

**Statistics and reproducibility**. Absolute counts were standardized according to relative abundance by dividing an ASV's count by the total number of counts within a sample as this has been reported to be more accurate than rarefying microbiome data[66]. Alpha diversity was estimated using the Shannon-Weaver index, taking the mean over replicates, followed by statistical testing of the diversity between environments by a one-way ANOVA and post-hoc Tukey's HSD test while correcting for multiple comparisons. Correlation between alpha diversity, depth, and DOC was quantified according to Pearson correlation and tested for significance using the Pearson's product moment correlation coefficient. Normality was checked prior to statistical testing with the Shapiro-Wilk test, Levene's test, and quantile-quantile plots. Beta diversity was estimated by the Bray-Curtis dissimilarity index and tested for significance between groups using a permutational analysis of variance (PERMANOVA). Prior to PERMANOVA, the homogeneity of within-group variation was assessed using PERMDISP[67]. The null hypothesis tested with this procedure was that the average within-group variation was equivalent among groups. Beta diversity was estimated according to Bray-Curtis dissimilarities and visualized using the nonmetric multidimensional scaling (NMDS) function on default settings in the R Vegan (v2.5)[68] package. Statistics and plot generation were performed in R Studio (version 3.6.3.)[69]. A compiled version of the R script, generated using knitr[70] (v1.33), is uploaded to a public repository with the link provided in the code availability statement below.

**Reporting summary**. Further information on research design is available in the Nature Research Reporting Summary linked to this article.

## Data availability
Data are available for the samples from nucleic acid sequencing repositories as detailed in Table S1.

## Code availability
The R Markdown document is provided in GitHub at: https://github.com/geweaa/connectivity/.

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

## Acknowledgements
The authors thank The Swedish Nuclear Fuel and Waste Management Co (SKB) for access to the Äspö HRL and Sicada database. The study was supported by The Swedish Research Council (contracts 2018-04311 and 2017-04422). M.D. thanks the Craafoord Foundation (contract 20180599). S.B. acknowledges financial support from the Swedish Research Council and Science for Life Laboratory. C.B. acknowledges funding by HIFMB, a collaboration between the Alfred-Wegener-Institute, Helmholtz-Center for Polar and Marine Research, and the Carl-von-Ossietzky University Oldenburg, initially funded by the Ministry for Science and Culture of Lower Saxony (MWK) and the Volkswagen Foundation through the "Niedersächsisches Vorab" grant program (grant number ZN3285). High-throughput sequencing was carried out at the National Genomics Infrastructure hosted by the Science for Life Laboratory. Bioinformatics analyses were carried out utilizing the Uppsala Multidisciplinary Center for Advanced Computational Science (UPPMAX) at Uppsala University (projects SNIC 2021/22-628 and SNIC 2021/6-256). The computations were enabled by resources provided by the Swedish National Infrastructure for Computing (SNIC) at UPPMAX partially funded by the Swedish Research Council through grant agreement no. 2016-07213.

## Author contributions
M.D. and S.B. conceived the study; M.D. and G.W. designed the research; G.W., S.T., M.M., L.A., V.S., C.B., J.P., M.K., M.Å., and S.B. produced and/or analyzed data; G.W., M.M., and M.D drafted the manuscript with comments from all authors.

## Funding

## Competing interests
The authors declare no competing interests.
