## [Transparent Peer Review File · Communications Biology]

Reviewers' comments:

Reviewer #1 (Remarks to the Author):

In the manuscript "Connectivity of terrestrial deep biosphere microbial biomass with surface communities" Westmeijer et al. explore the distribution and interconnectivity of microbial communities in surface and subsurface sediment and water samples using 16S rRNA sequencing. Taking advantage of the unique sampling opportunities in the ASPO HRL laboratories the authors collected samples from marine and benthic waters, shallow and deep marine sediments, terrestrial soil groundwaters, and deep meteoric and marine groundwaters. They evaluated the microbial diversity across these samples, looking for unique and shared ASVs, with the goal to identify in how far surface organisms infiltrate deep terrestrial groundwaters.

While I like the idea and approach of the authors and understand the motivation of the study, I have some general reservations on how the data is presented and how the paper is structured. I have no concerns with the language of the manuscript as it is mostly written coherently and grammatically correct.

Overall, I found the manuscript somewhat superficial and difficult to follow. Obviously, a great pool of data representing several different samples from unique locations was used for this study, however it was kind of difficult to understand which samples were taken and sequenced for this study and for which samples data existed. I know there is a lot information in the supplementary materials, but I urge the authors to include a large overview figure in the main manuscript, similar to the map in S1, but with more information. This will make it a lot easier for the reader to follow. The description of the results is also very shallow, the community compositions of the evaluated samples are only described very briefly, and while I understand that Patescibacteria may be the most important and interesting member I think a more detailed analysis and description of the communities would help to point out the key differences between the different, evaluated environments.

The authors also included a brief section on MAGs from a previous study, which they were able to associate with some of the most dominant ASVs in the here analyzed datasets based on an average nucleotide threshold of 90%. First, this analysis is described only very briefly in the methods section, omitting a lot of important information, second I am also a little skeptical to the threshold and how it was selected. If the authors would have wanted to investigate microbial function, I believe they should have included a metagenomic analysis of their samples. Otherwise, it would be better to exclude this add on analysis and really dig deeper into the 16S rRNA data.

Several of the addressed points are also highlighted by my smaller comments below:

Liner 42: I believe "limited" might be a better term here

Line 44: exhibit instead of show

Line 51: has shown instead of shows

Line 52: please remove "the" before sub-seafloor

Lines 52-53: This statement is phrased incoherently, what do you mean with "species sorting"? I assume you are trying to say that the geochemical conditions shape the microbial community drive the occurrence and abundance of distinct species?

Line 53: Please use "additional work" instead of "further study"

Line 53: "Additional work on Indian Ocean and up to 1.3 Ma old Bering Sea sediments obtained from

down to 332 m below the seafloor shows community..."

Line 56: Deep sediments?

Line 57: independent of the surface world

Line 63: chemistry

Line 69: Please rephrase, what do mean with cooperation? Synergism? Symbiosis?

Line 81: Such a map needs to be in the main manuscript, or at least some kind of overview figure

Lines 83-83: This is too little information, please provide some general numbers in the main manuscript. This will allow the reader to get a better idea of the data without having to look into the SI. For example, a range of sequencing reads or an average for each sample/environment.

Line 87: Some more detail on the Proteobacteria would be nice, what kind of Proteobacteria?

Line 93: What does "peaked" here mean? Please provide a number in the text.

Line 94: The same here, what was the lowest diversity?

Line 96: If you reference an earlier study here and specifically compare it to that study more information on that work needs to be provided.

Line 105: Beta diversity analysis showed

Line 110 -116: This is very confusing. Are you describing results from this study here? Or are you referencing previous work?

Line 129/130: Can you provide some more detail on these Deltaproteobacteria?

Line 160: This analysis needs to be described in more detail in the methods section. How was the 90% threshold selected?

Line 212: Please provide the depths here

Line 261: How was even sampling depth accounted for?

Comments to supplementals:

Fig. S1.: I think this map should be moved to the main manuscript, maybe more information on depth and sampling time could be added.

Fig.S2.: I understand the point of rarefaction curves, but it would be a lot more useful to have a table with the numbers.

Fig. S3: I suggest moving one of the lower level taxonomy plots to the SI. What cutoffs were used here, as the fraction "others" makes up more than 50% of the community for almost all of the sampled environments? Maybe more taxa should be depicted in the plot.

Fig. S4: That is a very high stress value

Reviewer #2 (Remarks to the Author):

Review comments to Westmeijer et al.

The manuscript entitled "Connectivity of terrestrial deep biosphere microbial biomes with surface communities" by Westmeijer et al. mainly employed 16S rRNA gene amplicon sequencing to assess the connectivity of microbiomes between the terrestrial groundwater environments and the surface communities. They also linked the dominant ASVs with draft genomes to assess the metabolic functions of the dominant microbes in the terrestrial deep biosphere at Äspö Hard Rock Laboratory. The presented analyses were based on an adequate number of samples with replicates to draw statistically significant conclusions. However, a number of important issues also deserve the authors' attention. For example, the study used a suite of samples from the Swedish coastal area near Äspö, but the study site was not mentioned in Abstract and Conclusion, and also many important statement sentences, which give the impression that the findings from Äspö are true for terrestrial deep biosphere everywhere. This study is not the end of the world, and the implications of this study should be explicitly stated for the terrestrial deep biosphere at Äspö or similar environments if there is no evidence showing that the deep biosphere at Äspö represents many other places. Also, another round of editions is needed to make the key messages clearly delivered in the opening or concluding sentences, especially in the Results and Discussion section. Below I first raised the major concerns and then some minor comments in detail.

Major comments:

Unlike marine sediments in which sample ages and connectivity can be easily interpreted based on the sample depths, physical connectivity of habitats in the terrestrial deep biosphere is unclear, at least is not clearly stated in the manuscript. The authors seem to assume that infiltration is the major force to drive the microbial migration and (literately) deep samples receive microbes from shallower samples, but there is no presented evidence to support it. Depending on tectonic histories, deep terrestrial samples can be younger; and depending on hydrological conditions, fluids can migrate from the deep to shallower depths (e.g., venting). So, physical connectivity between the habitats needs to be clearly established before biological connectivity can be discussed.

The samples were collected from three sites that are separated by hundreds of kilometers. In my opinion, some texts are required to justify that the surface microbes from separated sites (i.e., microbes in seawater collected from the Linnaeus Microbial Observatory and marine sediments from Loftahammar in this study) can represent those at Äspö. Does biogeography not matter in this area? This study uses amplicon sequencing datasets both generated by the authors and also obtained from previous studies. It is important to confirm whether the datasets were generated following the same lab procedure (e.g., DNA extraction, PCR amplification, and data processing), to minimize the bias introduced by potential different lab methodologies.

Lines 78-81: Some samples were classified as "aerobic", "suboxic", and "anoxic", while no O₂ concentration measurements were described throughout the manuscript. These sample features are important and were also mentioned in the text afterward (e.g., Line 102-104). Either direct measurements or reasonable speculations are required to justify these classifications.

Line 101 and Fig. 3: The "surface", "near-surface", "subsurface", "deep biosphere" environments need to be clearly defined. What samples were included in each of these types of environments?

Lines 107-109: "The shared ASVs ... comprise 98% of the old saline groundwater community" does not justify that "the old saline community was predominantly a selection of the modern marine groundwater community". If the water flow is from the old saline groundwater to the modern marine groundwater, then it can be that old saline groundwater contributes some microbes to the modern marine groundwater. So the key here is the water flow direction, without which it is hard to say which is selected from the other.

Line 79: why were there only two marine sediment layers included there, whereas 14 samples were included in Table S2? What samples were from 7 cm and 28 cm, respectively?

In Abstract, despite the hydrological connection with the Baltic Sea was mentioned, no particular terrestrial deep biosphere sites were specified, which gives the impression that the conclusion holds generally for all terrestrial deep biosphere sites. Explicitly spelling out the study sites would aid

readers to assess whether this is the case or not.

Lines 186-187: Patescibacteria are known to engage in episymbiosis with their prokaryotic hosts (see e.g., Castelle et al., (2018) Biosynthetic capacity, metabolic variety and unusual biology in the CPR and DPANN radiations. *Nature Reviews Microbiology* 16: 629-645). Could this facilitate the migration of them between different environments? Could their presence/thriving in particular environments be mainly controlled by their hosts rather than the bulk environmental conditions?

Minor comments:

Line 25 "such populations constituted up to 49% of the deep biosphere communities" reads contrasting with Line 27 "Most deep biosphere population lacked surface representatives, ..." It would be better to say clearly whether surface microbes constitute 49% of the total diversity or abundance of the deep biosphere.

Line 29: The phrase "persist by ..., anaerobic sulfur cycling" reads oddly. Shouldn't it be "persist by anaerobically cycling sulfur compounds"?

Line 63: "chemistry's" looks odd. It perhaps can be replaced by "chemical compositions".

Lines 96-100: The higher alpha-diversity observed in deep marine sediments than the surface sediment was attributed to the high sample numbers of the former than the latter (2 vs. 1). In my opinion, the conception that alpha-diversity of surface sediments is higher than those in deep sediments does not hold for every case. For example, some deep sediments in geochemical transition zones with local elevations of energy availability can sustain higher diversity of microbial communities than adjacent layers or even the most surface sediments (e.g., See Fig. 4 in Zhao et al. *Nitrifier abundance and diversity peak at deep redox transition zones. Scientific Reports* 9: 8633.). Could this be an alternative reason?

Lines 102-104: This sentence is hard to understand, which needs to be re-phrased. Also, to me it is unclear why the selection had already occurred in "the deep soil groundwater at 5 m". Why not the "aerobic shallow (2-3 m)"?

Lines 104-105: Why is this sentence here? What does "This" refer to? Also, is "Candidate Phyla Radiation" synonym to Patescibacteria? If no, examples should be listed for Candidate Phyla Radiation.

Lines 105-107: "a high differentiation" reads odd to me, because in my opinion differentiations were not quantified. Also, it is unclear what the sentence "the differentiation among ... was relatively low" means.

Lines 110-111: This is a very bold statement that the presented study alone is clearly not able to support. The term "photosynthesis fueled surface biomes" contains numerous biomes, most of which were not included in the study. How can the observed alpha diversity differences in the samples be generalized to "photosynthesis fueled surface biomes"?

Lines 182-184: References (either literature or figures/tables in this study) are needed to support this statement.

Figures

The sampling labels in Fig. 1 and Fig. S1 are really confusing. For each sample, please clearly spell out whether the sample is a "seawater", "groundwater", or "sediment". For example, the sample labeled "Marine benthic" in this study refers to the deep seawater environment but can be misunderstood as "marine benthic sediment". Also, "deep sediment" can be "deep marine sediment" or "deep freshwater sediment". Sample labels in Table S1 are more understandable, but still have some room for misunderstanding, which should be eliminated as much as possible.

Also, readers will greatly benefit from a two-dimension illustration showing the localities (different habitats) of the samples, which could be presented as another panel in Fig. 1. A map like Fig. S1 in my opinion is not enough to tell which is what, or where do the samples come from.

Also in Fig.1, why is only a single bar presented for each sample type, whereas multiple physical samples were collected and analyzed for each type?

Fig.2. It is important to emphasize that these samples do not represent a physical or chemical continuum, even though the vertical ordering of them following the decreasing depth trend.

Fig. 5. Only the completeness of the MAGs was given in Table S6. The contamination/redundancy level should also be included. Also, the details about how the genome size was estimated should be

described clearly.

Reviewer #3 (Remarks to the Author):

This manuscript presents analysis of a large dataset of 16S sequences obtained from different environments Baltic Sea coast in south-eastern Sweden . The key point of the authors is that surface microbes made up to 49% of the deeper subsurface communities, the rest of the community in deeper biosphere was not represented in the surface community, which are well adapted to subsurface deeper environments.

-When discussing deep biosphere, it is very difficult to ignore the archaeal population and focus only on bacterial communities. I see this as a shortcoming of this manuscript.

-Lack of accompanying geochemistry data, and correlation of ASV abundance and diversity based on these differences rather than only connectivity through water.

-'Deep sediment' should be defined better. How do you generalize 'deep sediment' results with samples from only 7 and 28cms? This is quite shallow, not even half a meter down. Sediment samples from different depths along the z axis will exhibit distinct differences in abundance and diversity of ASVs at depths all the way to 5m.

-Another concern I have is regarding the variable number of samples and therefore replicates from different sampling environment. While I can appreciate that for some of these it is challenging to obtain multiple samples, it should be logistically simpler to obtain multiple samples from say Shallow soil groundwater, Deep soil groundwater?

- The discussion on MAGs and SAGs is not organized and appears arbitrary. Were the MAGs assembled from metagenomes as part of this study? With ~ 60% completeness, there is only that much you can say. 6 belong to Patescibacteria, and some discussion on these, what about the other 24? The choice of pathways and genes highlighted for discussion is based on what? There is no discussion of genes that are likely to play an important role- motility, adhesion to surface, scavenging of metals for key enzyme activities, complex carbon metabolism etc- key traits for surviving in these environments.

Reference: COMMSBIO-21-1559A

Title: Connectivity of Fennoscandian Shield terrestrial deep biosphere microbiomes with surface communities

Kalmar, 28 October 2021

Dear reviewers,

Thank you for the constructive comments. We would hereby like to take the opportunity to submit a substantially revised version of the original manuscript. We have compiled the comments below and provided a detailed, point-by-point response.

We believe we have met the comments by *e.g.*, adding geochemistry data, improving the structure of the manuscript, and adding a section on physical connectivity between groundwaters. All comments were carefully considered and in parallel to the revised manuscript we provide an annotated version of the original manuscript to highlight all changes. Please note that the line numbers refer to the revised manuscript.

George Westmeijer *et al.*

Reviewer 1

In the manuscript “Connectivity of terrestrial deep biosphere microbial biomass with surface communities” Westmeijer et al. explore the distribution and interconnectivity of microbial communities in surface and subsurface sediment and water samples using 16S rRNA sequencing. Taking advantage of the unique sampling opportunities in the Äspö HRL laboratories the authors collected samples from marine and benthic waters, shallow and deep marine sediments, terrestrial soil groundwaters, and deep meteoric and marine groundwaters. They evaluated the microbial diversity across these samples, looking for unique and shared ASVs, with the goal to identify in how far surface organisms infiltrate deep terrestrial groundwaters. While I like the idea and approach of the authors and understand the motivation of the study, I have some general reservations on how the data is presented and how the paper is structured. I have no concerns with the language of the manuscript as it is mostly written coherently and grammatically correct. Overall, I found the manuscript somewhat superficial and difficult to follow. Obviously, a great pool of data representing several different samples from unique locations was used for this study, however it was kind of difficult to understand which samples were taken and sequenced for this study and for which samples data existed (1.1).

Reply to 1.1: We have clarified which samples were sequenced for this study and those previously published by adding this information to Fig. 2a and by adding the following sentence to **lines 114-116**: ‘The samples from the Baltic surface seawater, lower sediment, soil groundwaters and meteoric groundwaters were sequenced for this study while the data from the remaining environments have been previously published’. In addition, the samples previously published are described in the ‘Published datasets used in this study’ section starting at **line 253**.

I know there is a lot information in the supplementary materials, but I urge the authors to include a large overview figure in the main manuscript, similar to the map in S1, but with more information (1.2). This will make it a lot easier for the reader to follow.

Reply to 1.2: We thank the reviewer for the helpful suggestion and have moved the map from the supplemental materials to the manuscript as Fig. 2a and added whether the samples were sequenced for this study and those that were previously published (as requested in comment 1.1 above).

The description of the results is also very shallow, the community compositions of the evaluated samples are only described very briefly, and while I understand that Patescibacteria may be the most important and interesting member I think a more detailed analysis and description of the communities would help to point out the key differences between the different, evaluated environments (1.3).

Reply to 1.3: Substantial changes have been made in the manuscript to address this issue. The paragraphs ‘Microbial community description’ (**from line 126**) and ‘Connectivity of surface microbes with the deep biosphere’ (**starting on line 177**) were expanded, alpha diversity was correlated with depth (Fig. 3b) plus dissolved organic carbon (Fig. 3c), and groundwater chemistry (new Fig. 1) was included. Also, Figs. 5b and 5c were moved from the supplemental materials to the main manuscript.

The authors also included a brief section on MAGs from a previous study, which they were able to associate with some of the most dominant ASVs in the here analyzed datasets based on an average nucleotide threshold of 90%. First, this analysis is described only very briefly in the methods section, omitting a lot of important information, second I am also a little skeptical to the threshold and how it was selected. If the authors would have wanted to investigate microbial function, I believe they should have included a metagenomic analysis of their samples. Otherwise, it would be better to exclude this add on analysis and really dig deeper into the 16S rRNA data (1.4).

Reply to 1.4: As suggested, the analysis on microbial function has been excluded to focus more on community composition according to the 16S rRNA data.

Several of the addressed points are also highlighted by my smaller comments below:

Line 42: I believe “Limited” might be a better term here

Reply: Thank you, done

Line 44: exhibit instead of show

Reply: Done

Line 51: has shown instead of shows

Reply: Done

Line 52: please remove “the” before sub-seafloor

Reply: Done

Lines 52-53: This statement is phrased incoherently, what do you mean with “species sorting”? I assume you are trying to say that the geochemical conditions shape the microbial community drive the occurrence and abundance of distinct species?

Reply: We tried to phrase this more clearly: ‘In addition, analysis of marine sulfate reducing microbes has shown that the sediment surface community influences sub-seafloor populations through the process of species sorting whereby the geochemical conditions shape the microbial communities by favoring distinct populations’ (lines 54-56).

Line 53: Please use “additional work” instead of “further study”

Reply: Done

Line 53: “Additional work on Indian Ocean and up to 1.3 Ma old Bering Sea sediments obtained from down to 332 m below the seafloor shows community...”

Reply: Thank you, done.

Line 56: Deep sediments?

Reply: Done

Line 57: independent of the surface world

Reply: Done

Line 63: chemistry

Reply: Done

Line 69: Please rephrase, what do mean with cooperation? Synergism? Symbiosis?

Reply: Cooperation has been specified according to: ‘These microbial communities survive in the deep biosphere by syntrophy and symbiotic associations that alleviates the ‘tragedy of the commons’ that is aided by biofilm/aggregate formation’ (lines 76-78).

Line 81: Such a map needs to be in the main manuscript, or at least some kind of overview figure

Reply: Agreed, a map of the samples sites has now been added as Fig. 2a.

Lines 83-83: This is too little information, please provide some general numbers in the main manuscript. This will allow the reader to get a better idea of the data without having to look into the SI. For example, a range of sequencing reads or an average for each sample/environment.

Reply: The following text has been added to the manuscript: ‘In total 18 million sequencing reads were included in this study, encompassing 48.7 thousand amplicon sequence variants (ASVs). The upper sediment samples had the lowest sequencing depth with 34 ± 19 thousand reads on average per sample ($n = 19$) while the old saline groundwater samples had 108 ± 72 thousand reads ($n = 15$; details provided in Table S3)’ (lines 118-122).

Line 87: Some more detail on the Proteobacteria would be nice, what kind of Proteobacteria? Reply: Thank you for the suggestion. More detail has been provided: ‘Within the Proteobacteria, the Betaproteobacteriales

and the Campylobacterales orders were mainly responsible for the abundance of this phylum in the modern marine and old saline groundwaters (Fig. S3). On genus level, *Sulfurimonas* and *Thiobacillus* were abundant representatives of the Proteobacteria, together comprising 27% and 17% of the microbial abundance in the modern marine and old saline groundwaters, respectively' (lines 143-147).

Line 93: What does "peaked" here mean? Please provide a number in the text.

Reply: Done

Line 94: The same here, what was the lowest diversity?

Reply: Done

Line 96: If you reference an earlier study here and specifically compare it to that study more information on that work needs to be provided.

Reply: The comparison of the diversity decreasing with sediment depth was rephrased and extended. It now reads 'However, the lower sediment community alpha diversity was higher compared to the upper sediment (albeit insignificant, $p\text{-adj} = 0.22$) that contrasts with the general notion that diversity decreases with sediment depth^{27,48}. This incongruence was potentially caused by sampling a larger part of the sediment column (i.e. top 1 cm for upper sediment compared to 6 plus 20 cm depth for lower sediment), thereby capturing more fine-scale variation e.g., local diversity hotspots in redox transition zones⁴⁹, that would positively affect the overall diversity' (lines 154-159).

Line 105: Beta diversity analysis showed

Reply: Done, thank you.

Line 110 -116: This is very confusing. Are you describing results from this study here? Or are you referencing previous work?

Reply: Agreed, this paragraph has been rephrased to clarify where the results from this study are being reported and where a comparison is being made to a previously published study (lines 170-175).

Line 129/130: Can you provide some more detail on these Deltaproteobacteria?

Reply: More detail has been provided: 'In addition to the Patescibacteria, the Proteobacteria and Epsilonbacteraeota phyla were also abundant in the overlapping community (Fig 4B) and interestingly, the Epsilonbacteraeota phylum was only represented by the genera *Sulfurimonas* and *Sulfuricurvum*. The most abundant genera affiliated with the Proteobacteria were *Syntrophus* and *Hydrogenophaga*. Most Proteobacteria and Epsilonbacteraeota likely survived due to the prevalence and importance of sulfate reduction in oligotrophic deep biosphere groundwaters' (lines 192-197).

Line 160: This analysis needs to be described in more detail in the methods section. How was the 90% threshold selected?

Reply: The analysis on microbial function has been removed in response to comment 1.4 above.

Line 212: Please provide the depths here

Reply: Done

Line 261: How was even sampling depth accounted for?

Reply: The sampling depth was standardized for by using relative abundances instead of absolute counts and the following sentence was added to the manuscript to clarify this: 'Absolute counts were standardized according to relative abundance by dividing an ASV's count by the total number of counts within a sample as this has been reported to be more accurate than rarefying microbiome data' (lines 309-311). This is described in the following reference: doi:10.1371/ journal.pcbi.1003531.

Comments to supplementals:

Fig. S1: I think this map should be moved to the main manuscript, maybe more information on depth and

sampling time could be added.

Reply: Agreed, the map has been moved to the main manuscript as Fig. 2a.

Fig. S2: I understand the point of rarefaction curves, but it would be a lot more useful to have a table with the numbers.

Reply: The number of ASVs per sample is now included in the supplemental materials as Table S3 (former supplemental data file). The rarefaction curves are included to display that the majority of the samples had sufficient sequencing depth to justify diversity estimates. The following sentence was added to the manuscript to clarify this: ‘The rarefaction curves depicting the relationship between sequencing depth and ASV count are asymptotic for nearly all samples, indicating a sufficient sequencing effort for microbial diversity estimates (Fig. S2)’ (lines 122-124).

Fig. S3: I suggest moving one of the lower level taxonomy plots to the SI. What cutoffs were used here, as the fraction “others” makes up more than 50% of the community for almost all of the sampled environments? Maybe more taxa should be depicted in the plot.

Reply: The lower taxonomy plots (class, order) of the complete community are included in the supplemental materials as Fig. S3a (class) and S3b (order). In addition, the figure was changed and now shows the 19 most abundant groups. This cut-off was chosen as plotting the full diversity on *e.g.*, order level requires over 40 categories that makes the colors in the bar plot very difficult to distinguish.

Fig. S4: That is a very high stress value

Reply: We thank the reviewer for bringing this to our attention and unfortunately the stress value in the original Fig. S4 was incorrect and has been updated. The correct stress value of 0.06 has been corrected.

Reviewer 2

The manuscript entitled “Connectivity of terrestrial deep biosphere microbial biomes with surface communities” by Westmeijer et al. mainly employed 16S rRNA gene amplicon sequencing to assess the connectivity of microbiomes between the terrestrial groundwater environments and the surface communities. They also linked the dominant ASVs with draft genomes to assess the metabolic functions of the dominant microbes in the terrestrial deep biosphere at Äspö Hard Rock Laboratory. The presented analyses were based on an adequate number of samples with replicates to draw statistically significant conclusions. However, a number of important issues also deserve the authors’ attention. For example, the study used a suite of samples from the Swedish coastal area near Äspö, but the study site was not mentioned in Abstract and Conclusion, and also many important statement sentences, which give the impression that the findings from Äspö are true for terrestrial deep biosphere everywhere. This study is not the end of the world, and the implications of this study should be explicitly stated for the terrestrial deep biosphere at Äspö or similar environments if there is no evidence showing that the deep biosphere at Äspö represents many other places. Also, another round of editions is needed to make the key messages clearly delivered in the opening or concluding sentences, especially in the Results and Discussion section. Below I first raised the major concerns and then some minor comments in detail.

Reply: We agree that the study site at the Äspö Hard Rock Laboratory cannot be extrapolated to the entire terrestrial deep biosphere. However, based upon the study by Mehrshad et al. (please see <https://doi.org/10.1038/s41467-021-24549-z> that is also referenced in the manuscript), we argue that the data is relevant to the Fennoscandian Shield. Consequently, we have modified the title as well as adding both the ‘Äspö Hard Rock Laboratory, Sweden’ and ‘Fennoscandian Shield’ to the abstract (lines 27 and 33).

Major comments

Unlike marine sediments in which sample ages and connectivity can be easily interpreted based on the sample depths, physical connectivity of habitats in the terrestrial deep biosphere is unclear, at least is not clearly stated in the manuscript. The authors seem to assume that infiltration is the major force to drive the microbial migration and (literately) deep samples receive microbes from shallower samples, but there is no presented evidence to support it. Depending on tectonic histories, deep terrestrial samples can be younger; and depending on hydrological conditions, fluids can migrate from the deep to shallower depths (e.g., venting). So, physical connectivity between the habitats needs to be clearly established before biological connectivity can be discussed.

Reply: We agree with the reviewer and have added a paragraph ‘Geochemistry and water flow’ (lines 87-108) before discussing the microbial communities to discuss water flow between soil groundwaters and meteoric groundwater plus infiltration of modern marine groundwater by Baltic Sea water.

The samples were collected from three sites that are separated by hundreds of kilometers. In my opinion, some texts are required to justify that the surface microbes from separated sites (i.e., microbes in seawater collected from the Linnaeus Microbial Observatory and marine sediments from Loftahammar in this study) can represent those at Äspö. Does biogeography not matter in this area?

Reply: It has been demonstrated that salinity is the major driver for microbial community composition in the Baltic Sea (Herlemann *et al.* 2011 ISME J). Therefore, we have included salinity concentrations of all marine sites (LMO, Loftahammar, and Borholmsfjärden) and used statistical testing to show that no differences occurred between the sampling sites (lines 270-274).

This study uses amplicon sequencing datasets both generated by the authors and also obtained from previous studies. It is important to confirm whether the datasets were generated following the same lab procedure (e.g., DNA extraction, PCR amplification, and data processing), to minimize the bias introduced by potential different lab methodologies.

Reply: We agree with the reviewer and identical PCR primers plus settings, library preparation protocol, and bioinformatic processing were maintained for all samples. Regarding the DNA extraction, the protocol had to be adjusted to the nature of the sample, hence Qiagen’s PowerWater kit for groundwater samples and the

PowerSoil kit for sediment samples. For the Baltic surface seawater samples, taken at Linnaeus Microbial Observatory, a phenol-chloroform based method was used as described in Bunse *et al.* (2016) *Frontiers in Microbiology*.

Lines 78-81: Some samples were classified as “aerobic”, “suboxic”, and “anoxic”, while no O₂ concentration measurements were described throughout the manuscript. These sample features are important and were also mentioned in the text afterward (e.g., Line 102-104). Either direct measurements or reasonable speculations are required to justify these classifications.

Reply: Agreed. All classifications regarding oxygen concentration were removed from the manuscript as for most environments, direct oxygen measurements were missing. We attempted to use ferrous / ferric iron ratios as a proxy for redox state of the groundwaters. However, iron concentrations were often too low (below 0.10 mg L⁻¹) to allow the use of this ratio.

Line 101 and Fig. 3: The “surface”, “near-surface”, “subsurface”, “deep biosphere” environments need to be clearly defined. What samples were included in each of these types of environments?

Reply: To avoid confusion, the sentence has been rephrased to: ‘The ordination plot (Fig. 4) revealed a high dissimilarity between deep biosphere groundwater communities and Baltic surface and Baltic benthic, upper and lower soil, plus upper and lower sediment microbial communities, which was confirmed by statistical testing (Table S5)’ (lines 165-167).

Lines 107-109: “The shared ASVs ... comprise 98% of the old saline groundwater community” does not justify that “the old saline community was predominantly a selection of the modern marine groundwater community”. If the water flow is from the old saline groundwater to the modern marine groundwater, then it can be that old saline groundwater contributes some microbes to the modern marine groundwater. So the key here is the water flow direction, without which it is hard to say which is selected from the other.

Reply: We thank the reviewer for pointing this out. In response, the last part of the sentence: ‘the old saline community was predominantly a selection of the modern marine groundwater community’ has been removed and the text has been rephrased to: ‘The modern marine groundwater shared 2,186 out of its 9,315 ASVs with the old saline groundwater and these ASVs comprised 92% and 98% of the abundance in these groundwaters, respectively. This high degree of overlap between both communities is also depicted in Fig. 4 wherein the samples of both environments form a cluster. That the old saline groundwater community was predominantly a subset of the modern marine groundwater community suggested that its diversity was constrained by its geochemistry such as a very low organic carbon content (1 - 1.4 mg L⁻¹), long retention time (up to thousands of years), and very high chloride content (12 - 16 g L⁻¹)’ (lines 201-204).

Line 79: why were there only two marine sediment layers included there, whereas 14 samples were included in Table S2? What samples were from 7 cm and 28 cm, respectively?

Reply: Throughout the manuscript we refer to the sediment samples as ‘upper sediment’ that were taken from 0-1 cm sediment depth and ‘lower sediment’ that has been revised to 6 and 20 cm depths. Seven sites within Borholmsfjärden were sampled at two depths to account for fine-scale spatial heterogeneity in sediments giving a total of 14 samples. This has been clarified for the lower sediments on lines 267-270.

In Abstract, despite the hydrological connection with the Baltic Sea was mentioned, no particular terrestrial deep biosphere sites were specified, which gives the impression that the conclusion holds generally for all terrestrial deep biosphere sites. Explicitly spelling out the study sites would aid readers to assess whether this is the case or not.

Reply: The study site (Äspö Hard Rock Laboratory) was added to both the abstract (line 27) and conclusion (line 239).

*Lines 186-187: Patescibacteria are known to engage in episymbiosis with their prokaryotic hosts (see e.g., Castelle *et al.*, (2018) *Biosynthetic capacity, metabolic variety and unusual biology in the CPR and DPANN radiations. Nature Reviews Microbiology* 16: 629-645). Could this facilitate the migration of them between different environments? Could their presence/thriving in particular environments be mainly controlled by their*

hosts rather than the bulk environmental conditions?

Reply: Thank you for the suggestion. We have added this to the manuscript on lines 190-192.

Minor comments

Line 25 “such populations constituted up to 49% of the deep biosphere communities” reads contrasting with Line 27 “Most deep biosphere population lacked surface representatives, ...” It would be better to say clearly whether surface microbes constitute 49% of the total diversity or abundance of the deep biosphere.

Reply: Agreed. This has been changed to ‘... such populations constituted up to 49% of the deep biosphere microbial abundance’ (lines 27-28). This has also been changed for a similar sentence in the ‘Connectivity of surface microbes with the deep biosphere’ section (lines 181-188).

Line 29: The phrase “persist by ..., anaerobic sulfur cycling” reads oddly. Shouldn’t it be “persist by anaerobically cycling sulfur compounds”?

Reply: This sentence was removed when the analysis of microbial function section was deleted to focus on the community composition according to the 16S rRNA data (see reviewer 1 comment 1.4).

Line 63: “chemistry’s” looks odd. It perhaps can be replaced by “chemical compositions”.

Reply: This sentence has been rephrased as requested (lines 64-69).

Lines 96-100: The higher alpha-diversity observed in deep marine sediments than the surface sediment was attributed to the high sample numbers of the former than the latter (2 vs. 1). In my opinion, the conception that alpha-diversity of surface sediments is higher than those in deep sediments does not hold for every case. For example, some deep sediments in geochemical transition zones with local elevations of energy availability can sustain higher diversity of microbial communities than adjacent layers or even the most surface sediments (e.g., See Fig. 4 in Zhao et al. Nitrifier abundance and diversity peak at deep redox transition zones. *Scientific Reports* 9: 8633.). Could this be an alternative reason?

Reply: This discussion on diversity decreasing with sediment depth was rephrased and local diversity hotspots are included in the discussion (lines 156-162).

Lines 132-134: This sentence is hard to understand, which needs to be re-phrased. Also, to me it is unclear why the selection had already occurred in “the deep soil groundwater at 5 m”. Why not the “aerobic shallow (2-3 m)”?

Reply: This statement on selection was removed as discussing selection required additional metagenomic or metatranscriptomic data.

Lines 134-135: Why is this sentence here? What does “This” refer to? Also, is “Candidate Phyla Radiation” synonym to Patescibacteria? If no, examples should be listed for Candidate Phyla Radiation.

Reply: This sentence has been edited for clarity and moved up (lines 135-137) such that it is merged with the discussion of Patescibacteria abundance. It now reads 'The high abundance of the Patescibacteria clearly distinguished the groundwater from the sediment plus seawater communities (Fig. 2b) and is consistent with the identification of many Patescibacteria in shallow aquifers^{17,47,48}'.

Lines 135-137: “a high differentiation” reads odd to me, because in my opinion differentiations were not quantified. Also, it is unclear what the sentence “the differentiation among ... was relatively low” means.

Reply: The discussion on the beta diversity analysis was rephrased to: ‘The ordination plot (Fig. 4) revealed a high dissimilarity between deep biosphere groundwater communities and Baltic seawater, soil plus sediment microbial communities, confirmed by statistical testing (Table S5). The modern marine groundwater clusters with the old saline groundwater, while in contrast the meteoric groundwater sits alone between the other deep biosphere groundwaters and the lower soil groundwater’ (lines 165-169).

Lines 140-141: This is a very bold statement that the presented study alone is clearly not able to support. The term “photosynthesis fueled surface biomes” contains numerous biomes, most of which were not included in

the study. How can the observed alpha diversity differences in the samples be generalized to “photosynthesis fueled surface biomes”?

Reply: We agree with the reviewer and have toned down the statement such that it now reads: ‘The results showed that the deep biosphere groundwaters had a lower alpha diversity than the soil groundwaters and sediments. This diversity decreased with depth, retention time from a few years in meteoric groundwater to thousands of years in old saline groundwater, and dissolved organic carbon content (Fig. 3)’ (lines 170-172).

Lines 182-184: References (either literature or figures/tables in this study) are needed to support this statement.

Reply: This statement has been deleted when the analysis on microbial function was removed in response to reviewer 1 comment 1.4.

Figures

The sampling labels in Fig. 1 and Fig. S1 are really confusing. For each sample, please clearly spell out whether the sample is a “seawater”, “groundwater”, or “sediment”. For example, the sample labeled “Marine benthic” in this study refers to the deep seawater environment but can be misunderstood as “marine benthic sediment”. Also, “deep sediment” can be “deep marine sediment” or “deep freshwater sediment”. Sample labels in Table S1 are more understandable, but still have some room for misunderstanding, which should be eliminated as much as possible.

Reply: The references to the various environments (nine in total) have been edited for consistency and information and now read: Baltic surface seawater, Baltic benthic seawater, upper sediment, lower sediment, upper soil groundwater, lower soil groundwater, meteoric groundwater, modern marine groundwater, and old saline groundwater.

Also, readers will greatly benefit from a two-dimension illustration showing the localities (different habitats) of the samples, which could be presented as another panel in Fig. 1. A map like Fig. S1 in my opinion is not enough to tell which is what, or where do the samples come from.

Reply: The map in former Fig. S1 has been added to the main manuscript as panel ‘a’ in Fig. 2. We added the number of replicates within each environment and added if the data was generated for this study or was previously published.

Also in Fig.1, why is only a single bar presented for each sample type, whereas multiple physical samples were collected and analyzed for each type?

Reply: While we agree that it would be more correct to have a single bar for each biological sample, this would result in Fig. 2b having 100+ bars. Therefore, we added the number of replicates within each category to clarify that each bar represents multiple samples.

Fig. 2. It is important to emphasize that these samples do not represent a physical or chemical continuum, even though the vertical ordering of them following the decreasing depth trend.

Reply: We agree with the comment and have added the following sentence to the Figs. 2 and 3 legends: ‘The environments in panel b are ordered according to increasing depth although they do not represent a physical continuum due to the multiple sampling sites as shown in panel a.’

Fig. 5. Only the completeness of the MAGs was given in Table S6. The contamination/redundancy level should also be included. Also, the details about how the genome size was estimated should be described clearly.

Reply: This figure was removed when the microbial function section was removed in response to reviewer 1.

Reviewer 3

This manuscript presents analysis of a large dataset of 16S sequences obtained from different environments Baltic Sea coast in south-eastern Sweden. The key point of the authors is that surface microbes made up to 49% of the deeper subsurface communities, the rest of the community in deeper biosphere was not represented in the surface community, which are well adapted to subsurface deeper environments.

3.1 When discussing deep biosphere, it is very difficult to ignore the archaeal population and focus only on bacterial communities. I see this as a shortcoming of this manuscript.

Reply: While we agree that the 314F/805R primer pair is biased towards bacteria, we have previously shown that they amplify a range of archaea from the deep biosphere (e.g., please see doi: 10.1093/femsec/fiy121). In addition, we believe the focus on bacterial communities is justified as published metatranscriptomic studies in identical Äspö HRL deep biosphere groundwaters show that archaeal RNA transcripts consist of <1% of the total (please see: doi 10.1093/femsec/fiy121). Finally, ongoing unpublished data using archaeal specific primers demonstrate that archaeal diversity and abundance is low according to both 16S rRNA gene amplicon sequencing and qPCR.

3.2 Lack of accompanying geochemistry data, and correlation of ASV abundance and diversity based on these differences rather than only connectivity through water.

Reply: We agree with the reviewer and have added geochemistry data of the groundwaters as Fig. 1 and correlated alpha diversity with both depth and dissolved organic carbon (Fig. 3).

3.3 'Deep sediment' should be defined better. How do you generalize 'deep sediment' results with samples from only 7 and 28 cm? This is quite shallow, not even half a meter down. Sediment samples from different depths along the z axis will exhibit distinct differences in abundance and diversity of ASVs at depths all the way to 5m.

Reply: We have rephrased 'shallow' and 'deep' sediment to 'upper sediment' and 'lower sediment' for sediment depth of 0-1 and 6 plus 20 cm, respectively. Similarly, we refer to the two soil groundwaters as 'upper soil groundwater' and 'lower soil groundwater'.

3.4 Another concern I have is regarding the variable number of samples and therefore replicates from different sampling environment. While I can appreciate that for some of these it is challenging to obtain multiple samples, it should be logistically simpler to obtain multiple samples from say Shallow soil groundwater, Deep soil groundwater?

Reply: We agree that more samples from soil groundwaters would be desirable. However, we sampled all four soil tubes available on the Äspö Island, three for the shallow soil groundwater (2-3 m depth) and one for the lower soil groundwater (5 m depth).

3.5 The discussion on MAGs and SAGs is not organized and appears arbitrary. Were the MAGs assembled from metagenomes as part of this study? With 60% completeness, there is only that much you can say. 6 belong to Patescibacteria, and some discussion on these, what about the other 24? The choice of pathways and genes highlighted for discussion is based on what? There is no discussion of genes that are likely to play an important role- motility, adhesion to surface, scavenging of metals for key enzyme activities, complex carbon metabolism etc- key traits for surviving in these environments.

Reply: In accordance with the other reviewers, the analysis on microbial function was removed from the revised manuscript in order to focus on the connectivity between the various environments.

REVIEWERS' COMMENTS:

Reviewer #1 (Remarks to the Author):

In the manuscript "Connectivity of terrestrial deep biosphere microbial biomass with surface communities" Westmeijer et al. explore the distribution and interconnectivity of microbial communities in surface and subsurface sediment and water samples using 16S rRNA sequencing. Taking advantage of the unique sampling opportunities in the ASPO HRL laboratories the authors collected samples from marine and benthic waters, shallow and deep marine sediments, terrestrial soil groundwaters, and deep meteoric and marine groundwaters. They evaluated the microbial diversity across these samples, looking for unique and shared ASVs, with the goal to identify in how far surface organisms infiltrate deep terrestrial groundwaters.

This is a revised version of a manuscript I previously reviewed. While I already liked the concept of the study in my first review, I criticized the somewhat difficult to follow manuscript outline, as the authors had included various types of samples from different locations and it was often not clear which samples were analyzed as part of what project.

The authors responded well to my comments in the first round of reviews and were able to incorporate my suggestions and changes. I am quite happy with how the manuscript is structured in this revised version and I find it a lot easier to follow.

I have two minor points that I noted when reading through, which are technical. The authors state that all samples with reads below 1000 seq were discarded, which is fine, but did that result in the loss of a lot of data points? Related to this, you refer to the SI, but maybe it would be useful to mention the range or average number of reads per sample in the main text (only the total number of reads is listed).